# Antipsychotics result in more weight gain in antipsychotic naive patients than in patients after antipsychotic switch and weight gain is irrespective of psychiatric diagnosis: A meta-analysis

**Maarten Bak[1]\***, **Marjan Drukker[1]**, **Shauna Cortenraad[1]**, **Emma Vandenberk[1]**, **Sinan Guloksuz[1,2]**

1 Department of Psychiatry & Neuropsychology, Maastricht University, Maastricht, The Netherlands,
2 Department of Psychiatry, Yale University School of Medicine, New Haven, Connecticut, United States of America

\* m.bak@maastrichtuniversity.nl

## Abstract

### Introduction

Antipsychotics are associated with bodyweight gain and metabolic disturbance. Previous meta-analyses were limited to mainly antipsychotic switch studies in patients with a diagnosis of schizophrenia or psychosis with short follow-up periods. The present meta-analysis aimed to analyse the impact of weight change in antipsychotic-naive and antipsychotics switch patients and whether body weight change depended on diagnosis.

### Method

We performed a meta-analysis of clinical trials of antipsychotics that reported weight change, irrespective of psychiatric diagnosis. Outcome measure was body weight change. Studies were classified into antipsychotic-naive and antipsychotic-switch. Forest plots stratified by antipsychotic and the duration of antipsychotic use were generated and results were summarised in figures.

### Results

In total, 404 articles were included for the quantitative synthesis. 58 articles were on antipsychotic naive patients. In the antipsychotic naive group, all antipsychotics resulted in body weight gain. In the antipsychotic switch group, most antipsychotics likewise resulted in bodyweight gain, with exception of amisulpride, aripiprazole and ziprasidone that showed no body weight gain or even some weight loss after switching antipsychotics. Diagnosis was not a discriminating factor of antipsychotic induced weight change.

**Data Availability Statement:** All relevant data are within the paper and its Supporting information files.

**Funding:** None of the authors received any funding for the is study or receives funding that is conflicting with this study.

**Competing interests:** The authors have declared that no competing interests exist.

## Conclusion

Antipsychotic use resulted in substantial increase in body weight in antipsychotic-naive patients. In antipsychotic-switch patients the weight gain was mild and not present in amisulpride, aripiprazole and ziprasidone. In both groups, weight gain was irrespective of the psychiatric diagnosis.

## Introduction

Obesity is an increasing problem across the world and affects patients with a severe psychiatric illness disproportionately, leading to serious physical conditions such as diabetes mellitus type II, cardiovascular problems [1–3], cancer, infections, pulmonary problems, liver problems, mobility problems including lower back pain and arthrosis [4, 5], and comorbid mental health issues such as depression [6–8].

Several meta-analyses published during the last two decades show that antipsychotics (APs) result in weight gain, with clozapine and olanzapine associated with the most severe antipsychotic-induced weight gain (AIWG) [9–16]. Two recent network meta-analysis confirmed that weight gain in patients treated with clozapine and olanzapine significantly differs from weight gain in patients on placebo, whereas other medications (amisulpride, asenapine, brexpiprazole, chlorpromazine, haloperidol, iloperidone, paliperidone, risperidone, sertindole, quetiapine and zotepine) show no statistically significant AIWG compared with placebo [17, 18].

The differences in AIWG among APs are rather small and insignificant, except for olanzapine, when compared to metabolic friendly APs like amisulpride, aripiprazole and ziprasidone. Previous meta-analyses mainly analysed short term weight gain with a follow-up to a maximum of 13 weeks [17] or a median time of 6 weeks [18], However, APs are often used for longer periods. Furthermore, these meta-analyses only focus on patients with a diagnosis of schizophrenia and make no differentiation between AP-naive and AP-switch groups. We have previously shown that AP naive patients are more likely to develop more AIWG [11]. Furthermore, the majority of the randomised controlled trial (RCTs) included in the meta-analyses have been AP-switch studies, so far. Findings from AP-switch patients are different from findings in AP-naive patients because previous AP-treatment was also associated with weight gain [19]. The impact of APs on body weight change should therefore be ideally assessed in AP-naive patients to eliminate any influence of previous AP use.

Patients who received antipsychotic medication for the first time and had no history of antipsychotic use before, are defined as AP-naive patients. Patients who have ever used an antipsychotic before the study started and were randomised into an antipsychotic are called AP-switch patients. This is a heterogenous group as reasons for switching may vary: changing an antipsychotic because of lack of effect, because of weight gain or other side effects, because of the research protocol patients participate in, and other reasons.

Although metabolic effects of AP can be registered rapidly within weeks [19], body weight changes are predominantly longer lasting. Most patients with a severe mental illness (SMI) are in care for years and use AP for months to lifelong. In addition, AIWG seems to exceed the period of three months [14, 20]. Antipsychotics are not only prescribed to patients diagnosed with schizophrenia but also to patients with bipolar disorders, as well as off-label prescriptions. Some studies suggest that patients with schizophrenia are more at risk for developing weight gain and metabolic problems compared to patients with bipolar disorders or other diagnoses

[20–22]. However, others argue that diagnosis explains differences in weight change only because of the neuropharmacological properties of APs [2, 23]. Several neurotransmitter systems are responsible for the increase in appetite, decreased satiation, and thus increased food intake [24]. Antipsychotics display an array of interactions (through agonism or antagonism) of multiple neurotransmitters systems. Modes of action includes alterations in hypothalamic neuropeptides, striatal and mesolimbic dopamine reward pathways, gut hormones, peptides pituitary hormones, elevated proinflammatory cytokines, altered energy homeostasis and gut-dysbiosis [25]. Histamine 3, 5HT2c and 5HT1b, M1 and M3 are linked to increased food intake through increasing appetite [26, 27]. D2 antagonists, on the other hand, dampen the reward system urging the patient to eat more to experience some reward by food [28]. From a neuropharmacological view, neurotransmitters induced interactions of changes in appetite and metabolic disturbances must be irrespective of a psychiatric diagnosis. Consequences of the neurotransmitters induced interactions should therefore lead to the same effects in healthy volunteers [20]. The question is whether these more theoretical and experimental knowledge on AIWG is translated to clinical practice in daily life. Additionally, the psychiatric diagnosis remains the same during the study period and therefore does not affect changes in body weight in RCT's. Patients who gain body weight as a result of an antipsychotic may switch to another antipsychotic to stop weight gain or induce weight loss. In general, patients with a higher initial BMI may easier lose weight [29].

Previous (network) meta-analyses were, for the most part, restricted to shorter term studies, in mostly patients diagnosed with schizophrenia and did not make a clear distinction between AP-naive and AP-switch studies.

## Aim

An update of our previous meta-analysis that was restricted to studies from 1999 to 2012 has become necessary to include the clinical trials of new medications that were released after 2012 as well as all studies from 1960 onwards [11]. Further, as discussed above, there is a need to investigate AIWG 1- in AP-naive and AP-switch patients, separately, irrespective of duration of follow-up and 2- across diagnostic categories.

In addition, two more research questions were studied: 3- the difference in AIWG between AP naive and AP switch patients was analysed to evidence the separate analyses in both patients groups; 4- the difference in AIWG depending on average BMI at baseline was analysed.

## Method

### Data sources

A systematic review and meta-analysis of RCTs, controlled clinical trials, and randomised open label studies was performed; both blinded and open label long-term RCTs were included. The meta-analysis was conducted and reported according to the recommendations of the Meta-analysis of Observational Studies in Epidemiology (MOOSE) group [30] and the guidelines of the Preferred Reporting Items for Systematic Reviews and Meta-analyses (PRISMA) [31]. To ensure the MOOSE and PRISMA criteria were addressed and met, we followed a protocol intended for internal use and not published. This protocol was also used for the previous meta-analysis [11]. The Newcastle-Ottawa Scale (NOS) for assessing the quality of nonrandomised studies in meta-analysis search strategy, inclusion criteria and exclusion criteria were the same in previous meta-analysis. However, the present meta-analysis included studies with a publication date from 1960 and onwards (see below).

## Search strategy

A PubMed and Embase search was conducted to include research on body weight, body weight change, body weight gain in relation to antipsychotic medication published between 1-1-1960 and 30-06-2019. The search strings were provided in the S1 in S2 File.

Publications were screened by inspecting title and abstract to assess whether they met the inclusion criteria. Duplicates were removed. Abstracts or full texts were screened to exclude publications on rapid tranquillisation studies, reviews, meta-analyses, case reports, weight intervention studies, studies with duration of one week or brain morphology studies examining the effect of a single dose of medication. Subsequently, full text article screening resulted in exclusion of articles because of incomplete data, absence of crude data on body weight change, or failure to provide data per antipsychotic (when articles presented data as first generation antipsychotic (FGA) or second generation antipsychotic (SGA), these were treated as separate APs), overviews, risk assessment studies, case reports or papers with data redundancy. After qualitative assessment, 404 papers were selected and used for data extraction. During data extraction, papers that were extensions of a previous meta-analysis but with different baseline data or reporting a subgroup analysis were excluded (See Fig 1 Prisma Checklist flow diagram).

## Inclusion criteria and study evaluation

The aim of the search was to identify RCTs, controlled clinical trials, and randomised open label studies. The identified outcome was absolute change in weight. Studies were included if they reported data of one or more AP, or AP versus placebo or healthy control people. There were no restrictions with regard to diagnosis, age, dosage of AP or duration of AP exposure.

The inclusion criteria were:

1. Assessment of body weight change (continuous).

2. Age 18 years or older.

3. The data of weight change were available per AP.

4. RCT, controlled clinical trial or clinical trial or phase IV clinical trial with adequate control group with an intention to treat (ITT), open label studies with data per AP.

5. Publication date: 01-01-1960 / 30-06-2019.

## Exclusion criteria

- Studies which apply no ITT analysis.

- Very short term or acute antipsychotic interventions, rapid tranquilisation, or brain imaging studies conducted to assess the impact of AP on brain morphology or brain function were excluded. In these studies, antipsychotic interventions were very brief (ranging from a single dose to a 7-day regimen). These studies were excluded as they were not anticipated to show a clear change in body weight. Evaluation of weight change in short term interventions is often evaluated in the case of treatment of transient confusion or delirium which is complicated by underlying somatic illness that may explain body weight change directly and therefore represents a biased assessment of the impact of AP on weight change.

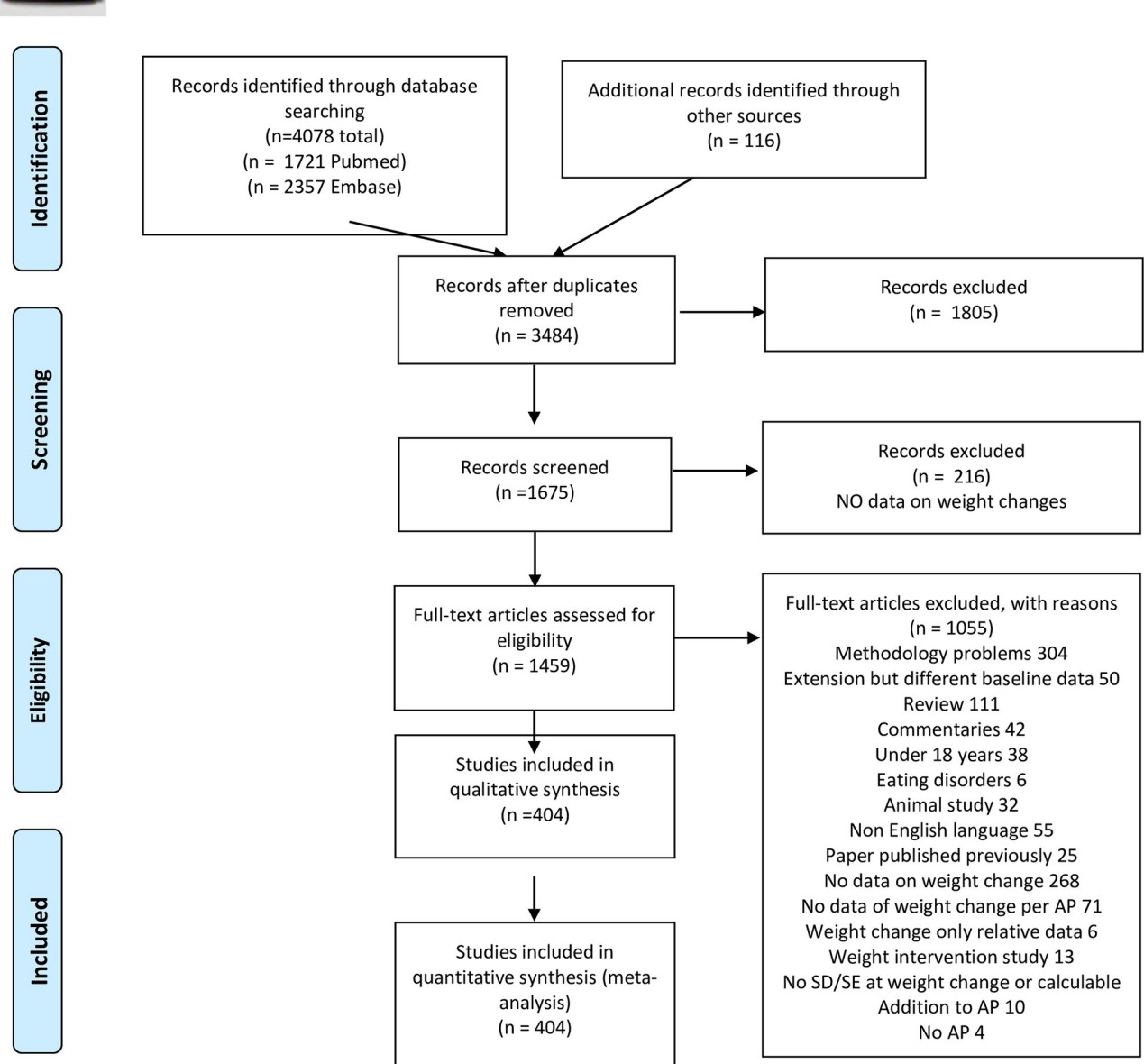

**Fig 1. PRISMA flow chart.**

- Studies designed to influence weight gain in patients with eating disorders such as anorexia or bulimia nervosa and studies involving somatic causes of weight change irrespective of the medication (e.g. delirium).

- Studies on specific (non-) pharmacologic interventions to reduce weight such as medication augmentation strategies, dietary programs, psycho-education or cognitive behavioural therapy (CBT).

- Systematic reviews, meta-analyses, case reports and poster presentations were also excluded. We also excluded retrospective cohort studies, or post-hoc reviews of clinical data and outcome of patients in care (either inpatients or outpatients.

- Studies reporting population with at least one subject <18 years of age.

- Non-human studies.

- Research published in languages other than English.

- Studies with weight changes given for the group of APs instead of individual APs.
(More detail information on exclusion after studying publication see S2 in S2 File)

No specific study protocol was made. This study was an extension of a previous meta-analysis [11]. All new articles (from 1/1/1960–12/31/1998 and from 1/1/2012 to 30/06/2020) were reviewed by two independent researchers (MB and SC or EV) who inspected the publications closely on the quality of the data presented in the studies, and outcome measures based on the PRISMA and MOOSE checklist criteria. We checked whether studies had a focused research question, a randomised study design, adequate and unbiased patient recruitment, unbiased measurement of outcomes, identification and control of major confounding factors, completeness of follow-up and accuracy of estimates.

In the case of conflict between reviewing researchers, publications were discussed with MB, SC and EV until consensus was reached. In case of remaining doubt the publication was further discussed with MD. The detailed evaluation and data entry were performed by SC and EV separately. MB supervised the search and data management process. New publications were entered into the existing database that was prepared for the previous meta-analysis [11]. All included research in previous meta-analysis were re-evaluated again independently by SC, EV and MB, to ensure they met the inclusion/exclusion criteria and qualitive standards of the current study inclusion and exclusion criteria as well as the methodological standard.

### Data extraction

Data from RCTs were extracted if they were based on ITT analysis. The non-ITT analysis publications were therefore not included. Only publications that reported body weight change per individual AP were included. The unit of weight was converted from pound to kilogram before data entry using an internet metric converting system (https://www.eenheden-omrekenen. info/eenhedenomrekenmachine.php?type=massa).

Duration was recoded into 4 categories (1–6 weeks, 6–16 weeks, 16–38 weeks and >38 weeks).

### Outcomes

The main outcome was defined as the body weight change in kilograms (kgs) after the initiation in the AP naive group or after the switch of an AP in the AP-switch group. Weight change was calculated by subtracting end of study body weight from baseline study body weight (body weight baseline—end body weight). In the instances in which standard errors were not

available, these were calculated using the formulas below:

$$sd\_change = sqrt(sd\_baseline^2 + sd\_endp^2 - 2 \times r \times (sd\_baseline) \times (sd\_endp))$$

$$se\_change = \left( {sd\_change} / {sqrt(n)} \right)$$

in which:

 r = correlation between weight at baseline and weight at follow up

 sd_change = estimated standard deviation of weight change scores

 sd_baseline = standard deviation of baseline weight

 sd_endp = standard deviation of endpoint weights

 se_change = estimated standard error of weight change

 n = number of subjects per study.

 r was estimated using data from a local longitudinal register of medication use in relation to somatic parameters [32] (data available July 2006–September 2012) as follows: weight change → 6–16 weeks: 0.96 (n = 220); 16–38 weeks: 0.95 (n = 241); 38–260 weeks 0.93 (n = 961); BMI change → 6–16 weeks: 0.96 (n = 212); 16–38 weeks (n = 240): 0.96; 38–260 weeks: 0.92 (n = 936). The r for duration of ≤ 6 weeks was also conservatively set at 0.96, as the longitudinal register had relatively few observations for this duration (n = 11) and in theory r increases when duration decreases.

## Number of antipsychotics included in statistical comparison

The number of antipsychotics complicates a comprehensive statistical analyses. Therefore, the number of AP included in the meta-analysis was restricted to those AP that were reported in the figures, i.e., information from at least 2 studies, which included the following APs: amisulpride, aripiprazole, brexpriprazole, haloperidol, lurasidone, olanzapine, paliperidone, risperidone, quetiapine and ziprasidone.

## Statistical analysis

All analyses were performed using Stata 16 [33]. In order to examine the outcome per antipsychotic for each duration of exposure category, the Stata command *metan* [34] generated forest plots including pooled estimates (absolute changes in kilograms) with their corresponding 95% confidence interval (95% CI). Analyses were stratified by AP-switch and AP-naive. A meta-regression was performed to test whether weight gain was different in AP- naive patients than in AP-switch patients (in each AP, separately).

 The computation of summary effects was carried out under the random-effects model, in which Tau was estimated using the DerSimonian-Laird method. Heterogeneity analyses were carried out using the chi-square, I-square, and Tau-square statistics. Tau-square estimates the total amount of variability (heterogeneity) among the effect sizes but does not differentiate between sources. Heterogeneity may be due to random or systematic differences between the estimated effect sizes. I-square estimates the proportion of the total variability in the effect size estimates that is due to heterogeneity among the true effects. $I^2 > 50\%$ was considered to indicate heterogeneity. We also presented figures per AP for each outcome measure. These figures were for descriptive purposes only.

 The variables diagnosis and naive/switch were added to the analyses to assess whether the association between AP and weight gain was different in different categories of the variables (moderation).

For the second and third aim, meta-regression analyses were performed. Finally, in 6 most frequently used AP and in the placebo groups, a meta-regression was performed with BMI at baseline as a modifier. These 6 AP and placebo also had sufficient data to assess publication bias stratified by naive/swich and duration as in the original analyses (with the exception of Clozapine in naive patients). Funnel plots were obtained and Egger tests were performed (Stata commands meta-funnel and meta-bias, respectively). Trim-and-fill analysis was performed to estimate effects of publication bias.

## Results

The search with Pubmed and EMBASE resulted in 1721 and 2357 titles, respectively. After checking for duplicates in both records and identifying publications through cross-referencing, we included 3484 publications (See Prisma flow diagram). As we built upon our previous meta-analysis, both papers included in the previous meta-analysis and papers obtained from the current Pubmed or Embase search were already in the database. Some papers were "additional records identified through other sources" Other papers were found through cross-referencing systematic reviews or meta-analysis or publications that were present in the data base of the previous meta-analysis but not in the current search. In total, we found 116 publications outside search strategies based upon Pubmed and Embase. Record screening by inspecting title resulted in 1675 papers eligible for abstract screening. Two hundred and sixteen papers had no data on body weight change, which resulted in 1459 papers for further independent screening and check on full-text eligibility (see Fig 1 PRISMA flow diagram). One publication was treated as two separate studies, as it presented two separate data sets in a single paper [35]. After checking for quantitative analysis, 404 studies eligible for analysis were included in the data base. For a more detailed explanation of the reasons for exclusion of publications in the meta-analysis, see PRISMA flow diagram (and S3 in S2 File: publication included in the study).

### Publication bias

Funnel plots and Egger tests showed some evidence for publication bias in some of the analyses (S7 and S8 in S2 File). In AP-switch patients, trim-and fill added studies, only in Olanzapine 6–16 weeks. In AP-naive patients trim and fill added studies in 16–36 weeks (Olanzapine) and >38 weeks (Olanzapine and Risperidone). In the rare case that trim-and-fill added studies, pooled estimate and p-value were rather similar to the original analysis (see also S7 and S8 in S2 File). Despite only the most frequently included AP were presented in a funnel plot, funnel plots with low numbers of studies could not be interpreted. Despite only the most frequently included AP were presented in a funnel plot, funnel plots with low numbers of studies could not be interpreted.

### AP-Naive weight changes

Only 72 publications reported data on AP naive patients (Table 1). All AP were associated with an increase in body weight since the start of AP treatment, with the only exception being paliperidone, which showed a weight gain over the shorter periods but no weight gain over the two longer periods. Placebo did not show any relevant weight change in all 4 periods. Only ziprasidone showed some weight gain in the longest period (See Fig 2). In most meta-analyses, heterogeneity was large, when 2 or more studies were included the I-square estimates were between 63 and 99.8 (one outlier the I-square was 13 aripiprazole < 6 weeks; Table 1).

Inspecting the figures for all antipsychotics in the AP-naive group, it was visible that the longer the AP use, the more the weight gain was. See the forest plots in supporting files for

**Table 1. Results AP-naive studies.**

| Antipsychotic | Time (wk) | N studies | n | Kg | 95%CI | I² | Tau² | Significance test Z | p |
|---|---|---|---|---|---|---|---|---|---|
| amisulpride | < 6 | 2 | 95 | 3.43 | 1.35–5.52 | 70.9 | 1.608 | 3.23 | 0.001 |
| aripiprazole | <6 | 4 | 311 | 0.28 | -0.26–0.82 | 13.3% | 0.048 | 1.00 | 0.315 |
| aripiprazole | 6–16 | 11 | 878 | 1.59 | 0.74–3.31 | 91.0% | 1.718 | 3.67 | 0.000 |
| aripiprazole | 16–38 | 2 | 129 | 2.39 | -1.73–6.51 | 99.3 | 8.755 | 1.14 | 0.255 |
| aripiprazole | >38 | 4 | 349 | 3.31 | 0.72–5.91 | 95.3% | 6.602 | 2.50 | 0.012 |
| clozapine | >38 | 3 | 207 | 6.21 | 2.31–10.18 | 92.5% | 10.864 | 3.12 | 0.002 |
| FGA | <6wk | 3 | 173 | 1.90 | 0.35–3.44 | 91.4% | 1.644 | 2.41 | 0.016 |
| FGA | 6-16wk | 4 | 226 | 2.88 | 0.57–5.20 | 94.5% | 5.215 | 2.44 | 0.015 |
| FGA | >38 | 6 | 234 | 7.97 | 6.41–9.53 | 88.6% | 2.692 | 10.02 | 0.000 |
| haloperidol | <6 | 3 | 173 | 1.90 | 0.35–3.44 | 91.4% | 1.644 | 2.41 | 0.016 |
| haloperidol | 6–16 | 4 | 226 | 2.88 | 0.57–5.20 | 94.5% | 5.215 | 2.44 | 0.015 |
| haloperidol | >38 | 6 | 234 | 7.97 | 6.41–9.53 | 88.6% | 2.692 | 10.02 | 0.000 |
| olanzapine | <6 | 24 | 1101 | 3.34 | 2.67–4.01 | 95.4% | 2.405 | 9.75 | 0.000 |
| olanzapine | 6–16 | 23 | 844 | 5.03 | 4.0–6.06 | 95.5% | 5.601 | 9.63 | 0.000 |
| olanzapine | 16–38 | 7 | 316 | 3.44 | 2.25–4.63 | 99.6% | 1.910 | 5.65 | 0.000 |
| olanzapine | >38 | 10 | 574 | 9.06 | 4.87–13.25 | 99.8% | 43.858 | 4.24 | 0.000 |
| paliperidone | 6–16 | 2 | 91 | 1.24 | -3.41–5.88 | 98.2% | 11.037 | 0.52 | 0.601 |
| quetiapine | <6 | 4 | 68 | 2.96 | 1.06–4.85 | 88.2% | 3.268 | 3.05 | 0.002 |
| quetiapine | 6–16 | 8 | 525 | 1.81 | 1.12–2.51 | 90.3% | 0.613 | 5.10 | 0.000 |
| quetiapine | 16–38 | 2 | 117 | 3.02 | -2.69–8.72 | 97.5% | 16.510 | 1.04 | 0.300 |
| quetiapine | >38 | 4 | 317 | 6.14 | 0.70–11.59 | 99.0% | 30.186 | 2.21 | 0.027 |
| risperidone | <6 | 6 | 284 | 3.35 | 2.11–4.60 | 95.1% | 2.078 | 5.29 | 0.000 |
| risperidone | 6–16 | 6 | 151 | 4.08 | 1.87–6.29 | 95.1% | 6.811 | 3.61 | 0.000 |
| risperidone | 16–38 | 5 | 194 | 5.19 | 1.35–9.03 | 99.0% | 18.438 | 2.65 | 0.008 |
| risperidone | >38 | 9 | 446 | 7.81 | 5.55–10.07 | 97.5% | 9.865 | 6.78 | 0.000 |
| sulpride | <6 | 2 | 24 | 0.43 | 0.08–0.78 | 96.8% | | 2.38 | 0.017 |
| ziprasidone | <6 | 4 | 204 | 0.37 | -0.73–1.46 | 90.9% | 1.018 | 0.65 | 0.513 |
| ziprasidone | 6–16 | 5 | 258 | 0.32 | -1.69–2.33 | 94.7% | 4.766 | 0.31 | 0.754 |
| ziprasidone | 16–38 | 2 | 89 | 0.50 | -6.16–7.16 | 96.8% | 22.378 | 0.15 | 0.883 |
| ziprasidone | >38 | 4 | 223 | 2.18 | -3.87–8.23 | 96.6% | 36.962 | 0.71 | 0.480 |
| placebo | <6 | 13 | 547 | 0.05 | -0.18–0.28 | 73.6% | 0.096 | 0.40 | 0.689 |
| placebo | 6–16 | 16 | 1135 | 0.33 | 0.14–0.53 | 98.4% | 0.054 | 3.34 | 0.001 |
| placebo | 16–38 | 3 | 218 | -0.91 | -2.05–0.24 | 98.4% | 0.976 | 1.55 | 0.122 |
| placebo | >38 | 3 | 120 | -0.12 | -2.42–2.18 | 63.9% | 2.399 | 0.10 | 0.917 |

Time is in weeks. N = number of studies. n = number of patient included in the study. Kg = change in kilogram body weight.

more detailed information per antipsychotic the forest plots in S4a in S2 File: Forest plots AP-naive studies on weight change. For some AP only 1 time period was identified. These results are presented in S9 in S2 File for both AP-naive as well as AP-switch.

## AP-switch group

Most studies were identified for the AP-switch group (N = 326). The most severe weight gain was found for clozapine in all 4 measurement periods: (<6wks) 3.29kg (95CI; 1.92–4.67), respectively (6-16wks) 3.77kg (95CI; 2.87–4.67), resp. (16-38wks) 3.94kg (95CI; 1.86–6.02) and resp. (>38wks) 10.9kg (95CI; 7.59–14.21), followed by olanzapine. Switching to

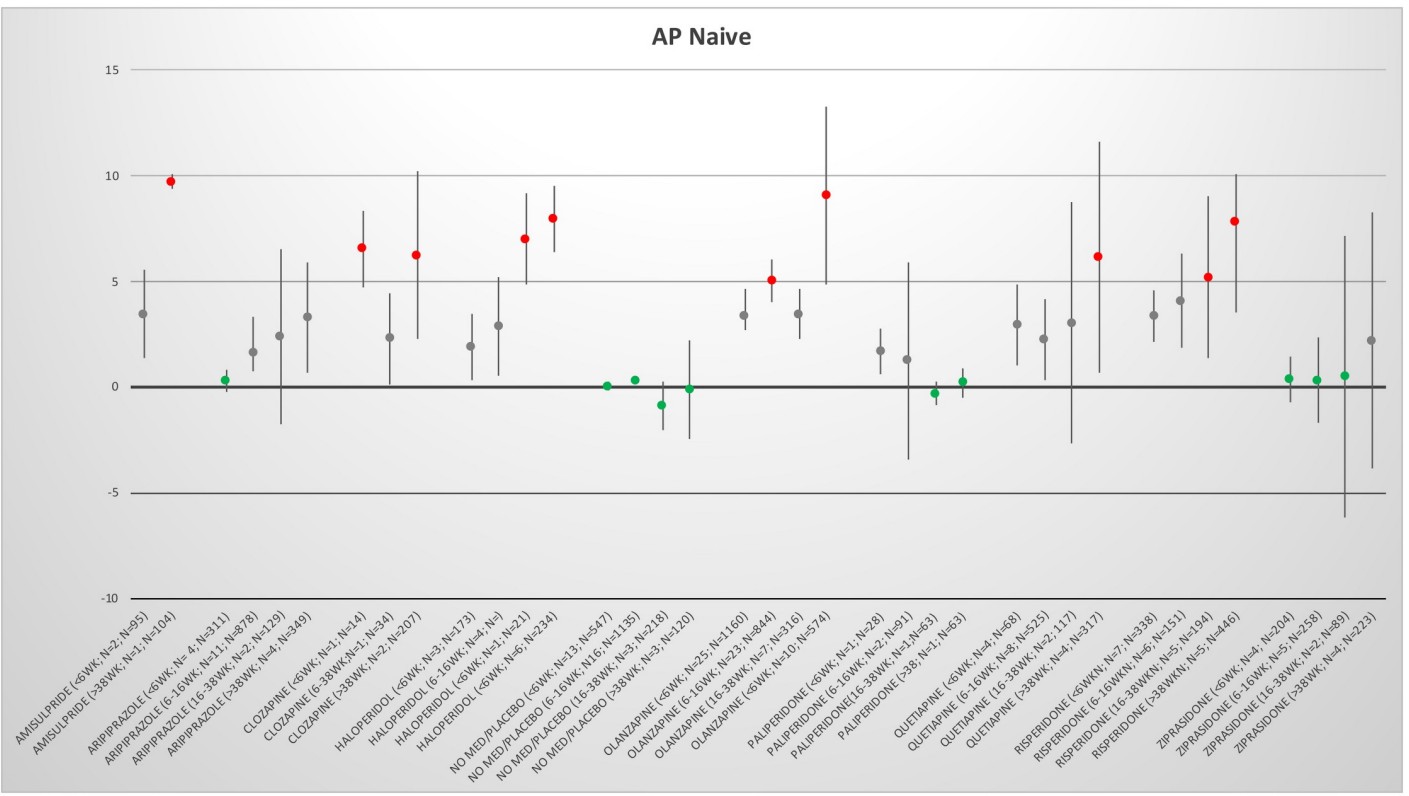

**Fig 2. Mean change in body weight per AP per period of the AP-naive group.** The antipsychotics per period. Green indicates almost no weight gain or weight loss. Grey indicates weight increase between 1 and 5 kilograms and red indicates weight increase > 5 kilograms per period. After the antipsychotic between brackets is indicated period in weeks, number of studies (N) and number of patients (n) were reported.

amisulpride, placebo and ziprasidone was associated with weight loss in the long term. Although switching to amisulpride was associated with a small increase in weight gain in the shorter terms, a decrease in body weight was observed in the longer period of 16–38 weeks: (16-38wks) -0.76kg (95CI; -2.74–1.22). After switching to ziprasidone, body weight decreased, and this loss of body weight was statistically significant in studies with longer duration (see Fig 3 and Table 2). Switching to placebo resulted in no or a small decrease in body weight See also Fig 3. Heterogeneity was large (the I-square was above 60 with the exceptions of 6 analyses).

All other antipsychotics were associated with body weight gain after AP-switch. Most APs were associated with a mild increase in body weight around 1–2 kgs at various time periods, but other antipsychotics resulted in more pronounced weight gain: chlorpromazine, FGA, SGA, and olanzapine (see Fig 3 and Table 2). See for more detailed information per antipsychotic the forest plots in S4b in S2 File: Forest plots AP-switch studies on weight change.

## AP-naive versus AP-switch

Placebo showed no difference in weight gain between AP-switch and AP-naive (B = 0.31;95% CI: -0.20–0.83). In amisulpride, aripiprazole, haloperidol, perphenazine, olanzapine,

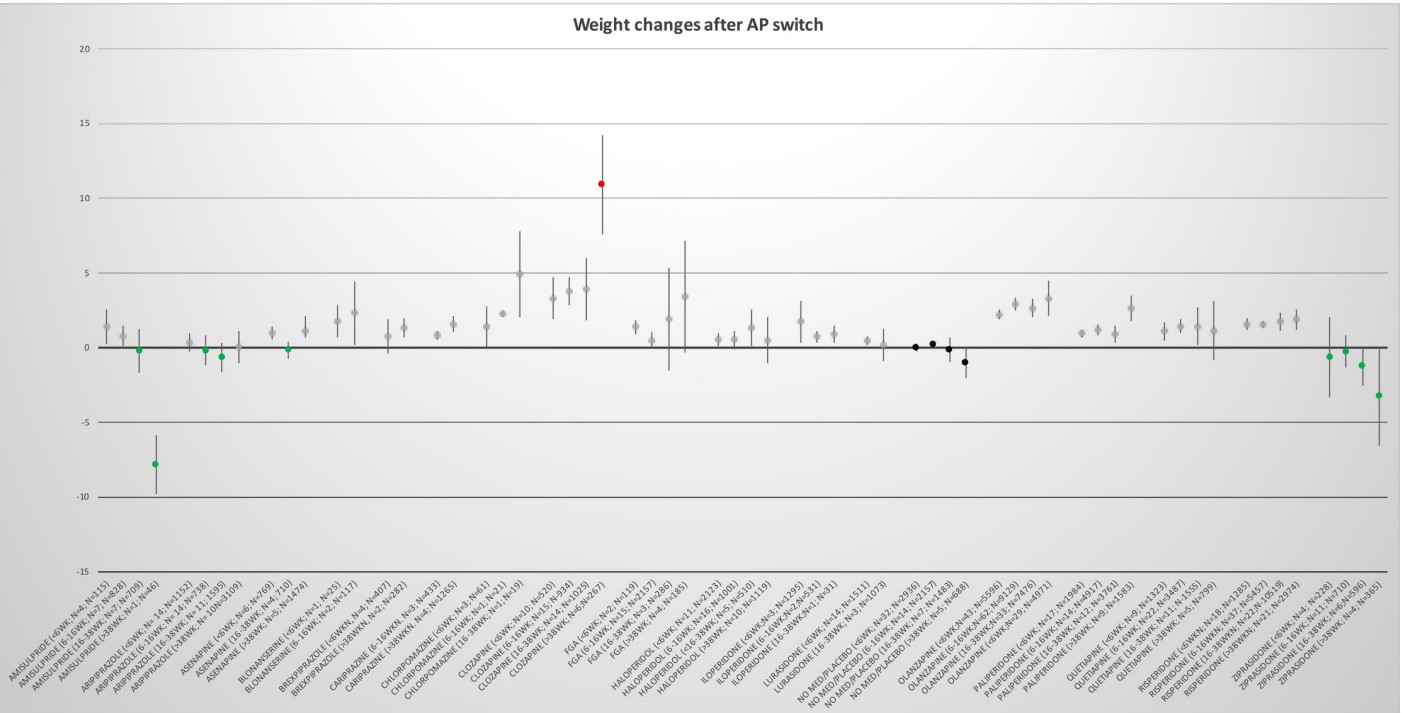

**Fig 3. Mean change in body weight per AP per period of the AP-switch group.** The antipsychotics per period. Green indicates almost no weight gain or weight loss. Grey indicates weight increase between 0 and 10 kg and red indicates weight increase > 10 kilograms per period. Black indicates placebo. After the antipsychotic between brackets is indicated period in weeks, number of studies (N) and number of patients (n).

quetiapine and ziprasidone, there was more weight gain in AP-naive patients than in switch patients (see Table 3). The remaining APs could not be analysed.

## Diagnosis

Analyses of differences in weight gain per diagnosis could only be performed in AP-switch patients. The AP-naive studies included almost only patients with a diagnosis of schizophrenia. Table 4 shows the meta-regression data. There was no statistically significant evidence that AIWG was dependent of psychiatric diagnosis (stratified by duration category, Table 4). However, for aripiprazole (6-16wk), olanzapine (>38wk), risperidone (<6wk) and ziprasidone (16-38wk), the effect sizes were high, indicating that AIWG and diagnosis might not be completely independent for all APs. (S5 in S2 File: All psychiatric diagnoses reported in the included papers).

## BMI at baseline as a modifier

In AP switch patients, some but not all strata of aripiprazole, clozapine, haloperidol and olanzapine showed that a higher initial BMI was associated with less weight gain (see S6 in S2 File). In AP-naive patients and in patients randomised to risperidone or placebo, this was not the case (see S6 in S2 File).

**Table 2. Results AP-switch studies.**

| Antipsychotic | Time (wk) | N studies | n | Kg | 95%CI | $I^2$ | $Tau^2$ | Significance test Z | p |
|---|---|---|---|---|---|---|---|---|---|
| amisulpride | < 6 | 4 | 115 | 1.40 | 0.23–2.57 | 72% | 1.022 | 2.34 | 0.019 |
| amisulpride | 6–16 | 7 | 828 | 0.78 | 0.11–1.45 | 93.9% | 0.646 | 2.29 | 0.022 |
| amisulpride | 16–38 | 7 | 709 | -0.20 | -1.68–1.28 | 95.4% | 2.577 | 0.75 | 0.452 |
| aripiprazole | <6 | 14 | 1152 | 0.34 | -0.27–0.95 | 99.0% | 1.094 | 1.09 | 0.274 |
| aripiprazole | 6–16 | 14 | 738 | -0.19 | -1.21–0.84 | 97.3% | 3.123 | 0.37 | 0.711 |
| aripiprazole | 16–38 | 11 | 1595 | -0.65 | -1.60–0.31 | 99.3% | 2.028 | 1.80 | 0.072 |
| aripiprazole | >38 | 10 | 3109 | 0.04 | -1.04–1.13 | 99.0% | 2.960 | 0.08 | 0.938 |
| asenpine | <6 | 6 | 796 | 0.96 | 0.55–1.38 | 82.0% | 0.227 | 4.44 | 0.000 |
| asenpine | 16–38 | 4 | 710 | -0.16 | -0.72–0.40 | 53.9% | 0.170 | 0.56 | 0.572 |
| asenpine | >38 | 5 | 1474 | 1.10 | 0.07–2.14 | 86.9% | 1.091 | 2.09 | 0.036 |
| blonanserine | 6–16 | 2 | 117 | 2.33 | 0.20–4.45 | 90.1% | 2.121 | 2.15 | 0.002 |
| brexipiprazole | <6 | 4 | 407 | 0.75 | -0.42–1.92 | 96.0% | 1.334 | 1.25 | 0.210 |
| brexipiprazole | >38 | 2 | 282 | 1.31 | 0.65–1.97 | 25.1% | 0.058 | 3.87 | 0.000 |
| cariprazine | 6–16 | 3 | 433 | 0.84 | 0.57–1.11 | 25.2% | 0.014 | 6.14 | 0.000 |
| cariprazine | >38 | 4 | 1265 | 1.57 | 1.04–2.11 | 64.7% | 0.188 | 5.73 | 0.000 |
| chlorpromazine | <6 | 3 | 61 | 1.38 | 0.01–2.74 | 89.4% | 1.302 | 1.97 | 0.049 |
| clozapine | <6 | 10 | 520 | 3.29 | 1.92–4.67 | 93.4% | 4.269 | 4.69 | 0.000 |
| clozapine | 6–16 | 15 | 934 | 3.77 | 2.87–4.67 | 84.5% | 2.405 | 8.18 | 0.000 |
| clozapine | 16–38 | 14 | 1025 | 3.94 | 1.86–6.02 | 98.3% | 15.301 | 3.71 | 0.000 |
| clozapine | >38 | 6 | 267 | 10.9 | 7.59–14.21 | 98.4% | 14.760 | 6.46 | 0.000 |
| haloperidol | <6 | 11 | 2123 | 0.53 | 0.09–0.98 | 91.1% | 0.472 | 2.36 | 0.018 |
| haloperidol | 6–16 | 16 | 1001 | 0.52 | -0.08–1.12 | 88.8% | 1.066 | 1.69 | 0.092 |
| haloperidol | 16–38 | 5 | 510 | 1.32 | 0.09–2.56 | 97.7% | 1.526 | 2.10 | 0.036 |
| haloperidol | >38 | 10 | 1119 | 0.48 | -1.06–2.03 | 94.4% | 5.344 | 0.61 | 0.541 |
| iloperidone | <6 | 3 | 1295 | 1.75 | 0.33–3.16 | 96.5% | 1.495 | 2.42 | 0.015 |
| iloperidone | 6–16 | 2 | 531 | 0.72 | 0.35–1.09 | 31.5% | 0.025 | 3.83 | 0.000 |
| lumateperone | <6 | 2 | 156 | | | | | | |
| lurasidone | <6 | 14 | 1511 | 0.46 | 0.20–0.72 | 75.1% | 0.179 | 3.47 | 0.001 |
| lurasidone | 16–38 | 3 | 1073 | 0.17 | -0.90–1.24 | 93.3% | 0.827 | 0.31 | 0.754 |
| olanzapine | <6 | 43 | 5596 | 2.22 | 1.92–2.51 | 91.5% | 0.773 | 4.72 | 0.000 |
| olanzapine | 6–16 | 62 | 9129 | 2.88 | 2.45–3.31 | 98.2% | 2.571 | 13.18 | 0.000 |
| olanzapine | 16–38 | 33 | 7476 | 2.64 | 2.04–3.24 | 98.8% | 2.802 | 8.62 | 0.000 |
| olanzapine | >38 | 29 | 4171 | 3.30 | 2.09–4.51 | 98.5% | 10.441 | 5.33 | 0.000 |
| paliperidone | <6 | 17 | 1984 | 0.96 | 0.71–1.21 | 76.8% | 0.204 | 7.48 | 0.000 |
| paliperidone | 6–16 | 14 | 4917 | 1.16 | 0.81–1.51 | 94.4% | 0.362 | 6.49 | 0.000 |
| paliperidone | 16–38 | 12 | 3761 | 0.88 | 0.32–1.44 | 89.3% | 0.808 | 3.05 | 0.002 |
| paliperidone | >38 | 9 | 1583 | 2.69 | 1.73–3.46 | 98.1% | 1.534 | 5.87 | 0.000 |
| quetiapine | <6 | 9 | 1323 | 1.10 | 0.48–1.72 | 93.9% | 0.695 | 3.50 | 0.000 |
| quetiapine | 6–16 | 22 | 3487 | 1.43 | 0.95–1.90 | 96.5% | 0.934 | 5.83 | 0.000 |
| quetiapine | 16–38 | 11 | 1555 | 1.43 | 0.21–2.66 | 98.8% | 3.557 | 2.30 | 0.021 |
| quetiapine | >38 | 5 | 799 | 1.15 | -0.83–3.13 | 93.3% | 3.983 | 1.60 | 0.109 |
| risperidone | <6 | 18 | 1285 | 1.57 | 1.17–1.96 | 82.2% | 0.579 | 7.51 | 0.000 |
| risperidone | 6–16 | 37 | 5457 | 1.54 | 1.31–1.76 | 93.9% | 0.328 | 13.44 | 0.000 |
| risperidone | 16–38 | 22 | 10519 | 1.75 | 1.13–2.37 | 99.0% | 1.834 | 5.54 | 0.000 |
| risperidone | >38 | 21 | 2974 | 1.89 | 1.19–2.56 | 88.7% | 2.037 | 5.38 | 0.000 |
| SGA | 6–16 | 2 | 80 | 6.69 | 5.51–7.87 | 89.7% | 0.646 | 11.15 | 0.000 |
| sertindole | 6–16 | 2 | 302 | 3.02 | 1.25–4.78 | 92.0% | 1.490 | 3.35 | 0.001 |

*(Continued)*

**Table 2.** (Continued)

| Antipsychotic | Time (wk) | N studies | n | Kg | 95%CI | I² | Tau² | Significance test Z | p |
|---|---|---|---|---|---|---|---|---|---|
| ziprasidone | <6 | 4 | 228 | -0.65 | -3.33–2.04 | 99.7% | 7.219 | 0.47 | 0.638 |
| ziprasidone | 6–16 | 11 | 710 | -0.25 | -1.33–0.83 | 96.8% | 2.882 | 0.45 | 0.651 |
| ziprasidone | 16–38 | 6 | 596 | -1.24 | -2.55 – -0.06 | 99.7% | 2.411 | 1.87 | 0.061 |
| ziprasidone | >38 | 4 | 365 | -3.25 | -6.55–0.05 | 99.7% | 10.736 | 1.93 | 0.054 |
| zotepine | <6 | 3 | 54 | 0.99 | 0.20–1.77 | 0.0% | 0.000 | 2.47 | 0.013 |
| placebo | <6 | 32 | 2936 | 0.03 | -0.27–0.21 | 95.3% | 0.36 | 0.25 | 0.799 |
| placebo | 6–16 | 14 | 2157 | 0.25 | 0.03–0.46 | 58.2% | 0.086 | 2.23 | 0.026 |
| placebo | 16–38 | 7 | 1483 | -0.12 | -0.96–0.71 | 95.3% | 1.150 | 0.29 | 0.772 |
| placebo | >38 | 5 | 688 | -1.01 | -2.02–0.01 | 89.9% | 1.182 | 1.96 | 0.050 |

Time is in weeks. N = number of studies. n = number of patient included in the study. Kg = change in kilogram body weight.

## Discussion

This meta-analysis is an update of a previous meta-analysis [11] and covers a longer period (1-Jan-1960 till 30-June 2019). Newer APs have been included as well and we have stratified by duration of AP treatment in studies using 4 time periods. In this update, AP-naive patients and AP switch patients were analysed separately. The main finding was that all APs were associated with mean weight gain over time, with the exception of ziprasidone. Placebo (a control compound) showed a small weight loss over time.

**Table 3. Meta regression analysis comparing body weight changes between AP-switch and AP-naive.**

| Medication | B | P | 95%CI | Heterogeneity I² |
|---|---|---|---|---|
| Amisulpride | 2.70 | 0.012 | 1.34–10.05 | 93.84% |
| Aripiprazole | 1.82 | <0.000 | 0.86–2.77 | 98.30% |
| Asenapine | NA | | | |
| Chlorpromazine | -0.04 | 0.979 | -3.42–3.34 | 82.52% |
| Clozapine | 0.71 | 0.659 | -2.52–3.95 | 97.74% |
| FGA | 3.85 | 0.148 | -1.46–9.16 | 93.53% |
| Haloperidol | 4.59 | <0.000 | 3.23–5.94 | 95.18% |
| Lurasidone | NA | | | |
| Paliperidone | -0.70 | 0.290 | -1.99–0.61 | 97.92% |
| Perphenazine | 2.41 | 0.09 | -0.94–5.75 | 00.00% |
| Olanzapine | 2.25 | <0.000 | 1.53–2.98 | 98.84% |
| Quetiapine | 1.71 | 0.003 | 0.58–2.83 | 98.23% |
| SGA | 6.70 | 0.064 | -0.56–13.94 | 97.72% |
| Sertindole | NA | | | |
| Ziprasidone | 1.83 | 0.028 | 0.20–3.46 | 99.15% |

NA = not applicable because there were no data available for AP naive weight change.

**Table 4. Meta-regressions weight change per diagnosis per period.**

| Antipsychotic | Period | Diagnosis | B | Confidence interval (95%CI) |
|---|---|---|---|---|
| | | Schizophrenia* | 1 | |
| aripiprazole | <6 wk | Bipolar disorder§ | -0.58 | -1.68–0.52 |
| | 6–16 wk | Bipolar disorder | 1.29 | -8.87–11.45 |
| | | Miscellaneous | 3.27 | -8.55–12.31 |
| | 16–38 wk | Bipolar disorder | 1.18 | -3.21–5.58 |
| | >38wk | Bipolar disorder | 0.78 | -1.63–2.35 |
| asenapine | <6wk | Bipolar disorder | 0.78 | -0.60–2.16 |
| haloperidol | <6wk | Bipolar disorder | 0.11 | -0.96–1.18 |
| | 6–16 wk | Bipolar disorder | 0.09 | -0.98–1.16 |
| lurasidone | <6wk | Bipolar disorder | -0.11 | -1.06–0.84 |
| olanzapine | <6wk | Bipolar disorder | -0.27 | -1.87–1.32 |
| | | Miscellaneous | -0.01 | -2.25–2.24 |
| | 6-16wk | Bipolar disorder | 0.37 | -1.28–2.01 |
| | | Miscellaneous | 0.22 | -2.55–2.99 |
| | 16–38 wk | Bipolar disorder | -1.50 | -5.26–2.27 |
| | >38wk | Bipolar disorder | -1.06 | -4.88–2.77 |
| | | Miscellaneous | -2.93 | -7.50–1.65 |
| paliperidone | <6wk | Bipolar disorder | 0.17 | -0.50–0.83 |
| | 6-16wk | Bipolar disorder | 0.59 | -0.20–1.38 |
| | 16-38wk | Bipolar disorder | -0.21 | -3.09–2.68 |
| quetiapine | <6wk | Psychosis other | -0.44 | -2.08–2.95 |
| | 6-16wk | Bipolar disorder | 0.37 | -2.13–1.40 |
| | | Miscellaneous | -1.26 | -2.69–0.16 |
| | 16-38wk | Bipolar disorder | -1.00 | -6.33–4.32 |
| | | Miscellaneous | -1.19 | -6.69–4.29 |
| risperidone | <6wk | Bipolar disorder | -8.85 | -1.91–0.21 |
| | 6-16wk | Bipolar disorder | -0.06 | -1.98–1.87 |
| | | Miscellaneous | 1.40 | -3.37–0.56 |
| | 16-38wk | Bipolar disorder | -0.53 | -3.82–2.77 |
| | >38wk | Bipolar disorder | -1.17 | -5.57–3.23 |
| | | Miscellaneous | -0.67 | -5.09–3.75 |
| ziprasidone | <6wk | Bipolar disorder | -1.36 | -4.65–1.94 |
| | 16-38wk | Miscellaneous | 2.29 | -3.53–8.12 |
| placebo | <6wk | Bipolar disorder | 0.21 | -3.32–0.74 |
| | | Miscellaneous | 0.53 | -0.83–1.90 |
| | 6-16wk | Bipolar disorder | 0.38 | -1.16–0.41 |
| | | Miscellaneous | -0.28 | -0.83–0.26 |
| | 16-38wk | Bipolar disorder | -0.55 | -3.42–2.32 |
| | >38wk | Bipolar disorder | 0.68 | -3.00–4.37 |

*Schizophrenia was the reference diagnosis. Other diagnoses groups were other psychotic disorders, bipolar disorder and miscellaneous group of all other psychiatric diagnoses).

§only outcome that could be analyzed were noted here.

## Difference in weight change between AP switch, AP-naive and baseline BMI

Meta-regression analysis showed that weight gain was larger in AP-naive patients in most APs (except paliperidone). Previous research also showed that the increase in body weight in the short term (first few months) was more noticeable in AP naive patients [36] than in AP switch patients. This finding is in line with the notion that the younger and more leaner patients are at risk the most [37, 38].

Nevertheless, the comparison between AP-naive and AP-switch also captures the notion that the AP-switch group is essentially a heterogenous group, as the reasons for switching antipsychotics are pluriform. These reasons for switching APs may include: lack of effect, as a result of weight gain or other side effects, because of the research protocol patients participate in and various other reasons. It might be expected that patients who gained a significant amount of body weight as a result of an AP treatment (e.g. olanzapine) may react with loss of body weight after switching to a more metabolically neutral AP. Despite switch groups are heterogeneous, the comparison between AP-naive and AP-switch shows that the differences in results between those two types of patients are important. Therefore it is necessary to analyse these groups of patients separately.

We hypothesized that patients with higher BMI at baseline would gain less weight during the study period. This means that patients with higher BMI, including those who gained a significant amount of body weight by a previous AP, are more likely to show weight decrease. Whereas lower BMI may increase the risk for more substantial body weight gain. This might explain why in the AP-naive show relative more weight gain, as they are found to be younger and more lean, and therefore more at risk for AIWG [37–39]. In this meta-analysis this cannot be tested. The analyses are just explorative, because BMI at the subject level was not available. Future original studies should be performed to replicate this finding.

## AP-naive

In AP-naive patients, body weight increased clearly over time for all antipsychotics, except for paliperidone. Strikingly, a serious increase of AIWG was also observed in aripiprazole, haloperidol, quetiapine and risperidone that are generally considered metabolic "friendly" APs [17, 18]. A meta-analysis using data from first episode psychosis patients showed a similar result [40], which replicates our previous study [11]. These findings indicate that these medications do result in serious weight gain in most patients with clinical implications. Patients with a younger age and lower BMI are more vulnerable for excessive weight gain [38, 39].

Earlier meta-analyses generally included shorter follow-up periods and predominantly AP-switch studies because of methodological pitfalls in studies with longer follow-up duration. The benefit in our current meta-analysis is that weight gain is examined across 4 stratified duration of AP use, which shows that body weight changes for a long time after the initiation or switch of an AP.

Recent meta-analyses did not stratify between AP-naive and AP switch patients, thereby underestimating the impact of antipsychotic medication on weight gain in AP-naive patients [9, 13, 17, 18]. Future RCTs and naturalistic follow-up studies are needed to clarify the real impact of APs on body weight in AP-naive patients over time, as weight gain is a predictor of non-compliance and metabolic and cardiovascular dysregulations [3, 22]. In addition, future RCTs need to analyse BMI and age at baseline as risk factors for AIWG over time.

## AP-switch

Previously, clozapine and olanzapine have been identified to be the two AP with most weight gain after AP-switch [9, 11, 17, 18, 41]. Previous meta-analyses found evidence that amisulpride, aripiprazole and ziprasidone are so-called metabolic "friendly" APs, suggesting that weight gain is minimal in AP-switch studies; and that an AP-switch might even result in weight loss [17, 18, 41]. In this meta-analysis, these three APs were not associated with clinically relevant weight loss after switch. The weight loss in amisulpride was restricted to only the last period of >38 weeks, which was based on data of one study only. Whereas the body weight loss of aripiprazole was minimal to nothing. Only ziprasidone showed a clear reduction in body weight after switching assessed at multiple time periods and weight loss was similar with placebo. The recently introduced APs, such as brexpiprazole and cariprazine, did not really differ from APs such as paliperidone or risperidone. Lurasidone is possibly more body weight "friendly" as results showed hardly any change in bodyweight. However, the number of studies and participants were limited. Recent network meta-analyses found that the impact of these AP on body weight were mild, with a modest body weight increase similar to aripiprazole after switching of an AP [17, 18]. Using network analysis in the present data may help understand whether lurasidone in the long-run results in significantly less weight gain than older APs.

## AIWG and diagnosis

The present meta-analysis did not provide evidence that diagnosis is a moderator. This means that the AP related body weight changes are irrespective of the diagnosis. This seems counter-intuitive. But the research literature is inconclusive. Some studies conclude that patients with schizophrenia appear more at risk for metabolic syndrome and diabetes mellitus compared with patient with bipolar disorder [42]. Whereas others reported no evidence for differences in body weight change depending on diagnosis [43–45]. The finding that diagnosis was not associated with AIWG emphasizes the impact of neurotransmitters involved, steering neurobiological experiences like feeling hungry related to 5HT2c or Histamine antagonism. Feeling hungry likely results in carbohydrate intake, especially if the reward system is blocked by the D2 antagonist. Similar to every healthy person, patients are very tempted to eat, especially hedonic foods like carbohydrates and fats [20]. This is unrelated to any psychiatric diagnosis but foremost a normal behaviour given the neuropharmacological changes that are the result of APs' mechanisms of action [46].

## Duration of AP use

When visually inspecting Fig 2, it can be observed that the longer an AP is used, the more weight gain is noted. Hallmark meta-analyses did not control for the duration of an AP as a contributing factor for weight gain. In these meta-analyses, either the results were re-calculated towards a 10 weeks period follow-up period, assuming that weight gain is a linear process (e.g. for medication X, if the weight gain reported was 4 kgs in 20 weeks, and consequently weight in 10 weeks was approximated as 2 kgs) [9], or only studies with data in a specific period (e.g. 4–12 or 3–13 weeks) were included [13, 14, 17] or data closest to a duration of 6 weeks were included (e.g. median treatment period of 6 weeks) [18]. Long-term studies are more prone to methodological flaws. In addition, when follow-up duration increases, the direct association between weight gain and the AP-medication weakens gradually.

Our findings reiterate that the monitoring and management of bodyweight from the beginning of AP treatment is extremely important since weight gain is easy to measure and follow-up; and weight loss interventions offer direct improvement on physical and mental

functioning. Focussing on loss of body weight results in reduction of risk for most of the somatic problems related to being overweight or obesity like Diabetes Mellitus type II or cardiovascular diseases [47]. Interventions targeted on body weight loss appear feasible in people diagnosed with severe mental illness [48, 49]. Monitoring and body weight management at the beginning of treatment with an AP are important. Ultimately, prevention of weight gain is many times preferable than painful efforts to lose body weight.

## Limitations

This comprehensive meta-analysis addresses an important topic and includes 404 papers on AP-induced body weight changes that includes all studies irrespective of the study duration and psychiatric diagnoses. However, the present meta-analysis has also some limitations.

First, in the AP-naive group, many studies were excluded, because these studies did not meet the inclusion criteria age >17 years. This might have resulted in an underestimation of the weight gain as the younger aged individuals with lower BMIs are more vulnerable for excessive weight gain [39]. Many first episode studies or studies in AP-naive patients were conducted in youth populations. The rationale of excluding these patients, hence these studies, is that young patients also gain weight as a result of the natural growth and maturing. Therefore, including these studies with adolescents could have biased results. A meta-analysis including only studies in adolescents can provide valuable information, but studies that control for normal growth need to be included. Also, the use of BMI rather than body weight is not sufficient to tackle this challenge, because BMI in healthy adolescents also increases with age.

Second, the number of AP-naive patients was rather small for some AP. Therefore, the results need to be interpreted with caution. However, the outcomes of the most prescribed drugs were robust and all findings point in a same direction. Nevertheless, more studies in AP-naive patients are needed. For the new APs like brexpiprazole, cariprasine and lurasidone, no studies in AP-naive patients could be included. Therefore, there exists no information whether these drugs are associated with weight gain in AP-naive patients.

Third, controlling for confounders like dosage and ethnicity was not possible in a meta-analysis when the original study did not report this. When original studies only provided characteristics at the group level, individual differences could not be analysed.

We included English written papers only, because we otherwise would have trouble understanding and judging the papers for methodological and outcome properties. Therefore, a few studies from China and other Asian countries were excluded. Patients in Asian countries are found to have a lower BMI and respond with lesser weight increase compared to those Western countries [40]. In another recent meta-analysis, Chinese studies were excluded due to a higher risk of methodological shortcomings [17]. In the present meta-analysis, we included studies from China that were published in English as we were able to evaluate and confirm that the studies were methodologically sound. The included research from China that were conducted in China authored by Chinese authors solely were all published in peer-reviewed journals, thereby ensuring a level of scientific validity and reliability. In the AP naive and AP switch studies, the number of studies with a pure Asian population were limited and not expected to influence the data or lead to slight underestimation of the AIWG.

Furthermore, some studies did not present Sd (standard deviation) or Se (standard error) of weight change; and Se could also not be calculated from Se's at baseline and follow-up. When Se was missing, mean changes could not be analysed. Contacting authors of studies from 2008 or more recent studies resulted in some extra data input, but the majority of authors did not respond. In this respect, we recommend that future RCTs report data of

adverse effects (i.e. weight gain) according to standards and include measures of variability such as Sd, Se or 95% confidence interval.

Finally, unpublished work was not included in this meta-analysis. We checked for poster presentations as this might overcome this partly. Studies with null-findings are more likely to remain unpublished. Because of the limited number of studies in various AP, controlling for publication bias was not possible in all AP. All results remained the same after missing studies were added all results remained the same. This indicates that for olanzapine and risperidone publication bias was not of great importance. It is plausible to speculate whether publication bias is likely similar in the AP with smaller numbers of studies.

## Conclusion

All antipsychotics result in weight gain except for ziprasidone in AP-switch studies. Differences between AP were not tested as we could not conduct a network meta-analysis, given the multi-layered structure of the data. In contrast to our previous meta-analysis, we have now stratified studies into AP-naive and AP switch groups. The AP-naive AIWG was noticeable for all antipsychotics in short and longer terms. This finding calls for immediate follow-up and prevention programmes. Switching to a metabolic friendly AP in the case of AIWG does not necessarily result in weight loss. Troublesome is the most effective AP clozapine and olanzapine also show the most severe increases in body weight. If necessary, switching to a less hazardous AP such as aripiprazole, amisulpride and ziprasidone might be recommended. Lurasidone also appears to result in less weight gain, but more studies are required to confirm this. However, considering all effects of an AP before initiating or changing an AP is crucial. Therefore, before initiation an AP in AP-naive patients take all effects and risk factors for increased liability of adverse effects like weight gain into consideration. In contrast to guidelines that recommend AP-switch as a first step to deal with AP-related body weight increase, we would strongly advise a critical evaluation of alternative strategies for losing body weight, such as individual lifestyle counselling and exercise interventions before switching AP [50, 51].

## Supporting information

**S1 File.**
(DOC)

**S2 File.**
(DOCX)

**S3 File.**
(XLSX)

## Author Contributions

**Conceptualization:** Maarten Bak, Marjan Drukker, Sinan Guloksuz.

**Data curation:** Maarten Bak, Shauna Cortenraad, Emma Vandenberk.

**Formal analysis:** Marjan Drukker.

**Investigation:** Maarten Bak, Shauna Cortenraad, Emma Vandenberk.

**Methodology:** Maarten Bak, Marjan Drukker, Sinan Guloksuz.

**Project administration:** Maarten Bak.

**Supervision:** Maarten Bak, Marjan Drukker, Sinan Guloksuz.

**Validation:** Maarten Bak, Marjan Drukker, Sinan Guloksuz.

**Visualization:** Maarten Bak.

**Writing – original draft:** Maarten Bak, Marjan Drukker.

**Writing – review & editing:** Shauna Cortenraad, Emma Vandenberk, Sinan Guloksuz.

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
