## [Decision Letter · Decision Letter 0]

12 Oct 2020

PONE-D-20-24708

Antipsychotics results in more weight gain in antipsychotic naive patients compared to antipsychotic switch and weight gain is irrespective of psychiatric diagnosis

PLOS ONE

Dear Dr. Bak,

Thank you for you patience in the review of your manuscript. We worked hard to obtain qualified reviewers and some reviewers requested additional review time which we granted. 

Thank you for submitting your manuscript to PLOS ONE. After careful consideration, we feel that it has merit but does not fully meet PLOS ONE’s publication criteria as it currently stands. Therefore, we invite you to submit a revised version of the manuscript that addresses the points raised during the review process.

We look forward to receiving your revised manuscript.

Kind regards,

Kyle J Burghardt

Academic Editor

PLOS ONE

Additional Editor Comments:

Please check your manuscript carefully for English-language grammar and editing.

Journal Requirements:

2. Please note that PLOS ONE requires that systematic reviews and meta-analyses are labeled as such in the title of the manuscript. we ask that you revise your title according. I.e., "Antipsychotics result in more weight gain in antipsychotic naive patients compared to antipsychotic switch, and weight gain is irrespective of psychiatric diagnosis: a meta-analysis

4. Please include your tables as part of your main manuscript and remove the individual files. Please note that supplementary tables (should remain/ be uploaded) as separate "supporting information" files.

Reviewers' comments:

Reviewer's Responses to Questions

**Comments to the Author**

1. Is the manuscript technically sound, and do the data support the conclusions?

Reviewer #1: Yes

Reviewer #2: Yes

2. Has the statistical analysis been performed appropriately and rigorously? 

Reviewer #1: Yes

Reviewer #2: Yes

3. Have the authors made all data underlying the findings in their manuscript fully available?

Reviewer #1: Yes

Reviewer #2: Yes

4. Is the manuscript presented in an intelligible fashion and written in standard English?

Reviewer #1: Yes

Reviewer #2: No

5. Review Comments to the Author

Reviewer #1: This paper is an update of an earlier meta-analyses (2014) of this group, and findings are generally the same i.e. that almost all AP's are associated with weight gain. The authors take a longer period of finding studies (from 1960 onwards) and claim they look at longer follow up periods, but they did the latter also in their previous meta-analysis. In the 2014 meta-analysis they state most studies were AP switch studies while for the current one they assert 'not only AP-naive patients but also AP switch patients were analysed, separately.' This does not correspond well. Apart from including some of the newer APs and the comparison of switch and naive patients, not much is new in this paper. No network meta-analysis is performed (while several of such meta-analyses have been published some years ago).

Having said that, I don't really understand the comparison between naive and switch patients. They are so completely different patient groups. In the switch group, patients can be included that insufficiently benefit from current AP and may need to switch to APs like olanzapine and clozapine, which are associated with considerable weight gain. It can also include patients that have already gained weight and want to try a 'metabolic friendly' AP like aripiprazole. When patients are already obese, it is extremely difficult to loose weight. In contrast, the naive group will be much younger, the AP use might be for the first episode psychosis. The naive patients will very likely increase in weight, the switch group will show a mixed picture, depending on the reason for a switch. To me it seems odd to compare these two groups. Maybe the authors could state in the aims more clearly why it is interesting to compare these groups. Of course, results from both groups (separately) are relevant.

Other remarks:

* I don't know whether I agree that most meta-analyses are on switch patients; some of the more recent meta-analyses that I can find are of patients with acute psychotic symptoms.

* the search in PubMed and EMBASE only seems limited. There are so many more literature data bases (e.g. PsychInfo, Cochrane, CiNAHL). Furthermore, authors state they fear for publication bias as null results are less likely to be published, but have they checked the trial registries such as ClinicalTrials.gov, to search for trials that have been carried out but are not published? That would increase the reliability of the results.

* Some of the important information on e.g. publication bias, is presented only in the discussion and not in results. Also other posthoc analyses i.e. the role of BMI at baseline, are newly presented in the discussion while I think these belong earlier in the paper.

* on page 5, it is asserted that from the neuropharmacological view neurotransmitter induced interactions are independent of psychiatric diagnoses. This standpoint deserves some explaining.

* on page 6 (and 7 and 8), authors contradict themselves in stating that they include RCTs only but also look at cohort studies, nonrandomized and other types of non-controlled studies. It should be made clear what types of studies authors include. (On page 9 it is mentioned again that authors looked for randomized studies.)

* On page 12, authors write: "For the second aim, meta-regression analyses were performed to assess whether diagnosis was a modifier." This could be described in more detail as not everyone will be acquainted with meta-regression analysis.

* On page 14, figure 3, it is mentioned that number of patients is also in the figure (just like in figure 2) but this is not the case. Moreover, for amilsupride, the number of studies, for which long term effects (>38 weeks) were analysed, is not mentioned.

* In figure 2, for haloperidol, the periods are not formulated correctly (twice <6 wks), the number of patients also seem to be missing. The reporting does not seem very concise.

* on page 16, first paragraph of the discussion, the last sentence is: "The main finding was that all APs were associated with mean weight gain over time, with the exception of ziprasidone and placebo." However, "placebo" is not an antipsychotic.

* page 17, first paragraph, the last sentence is repetition, this same sentence is also the last sentence on page 16.

* Page 19, top paragraph, the first sentence starts with 'lifestyle aspects', and then the second sentence starts with "This contributes". It is unclear to what 'this' is referring to. More in general, this paragraph about lifestyle, weight gain, diet, interventions and implication is not very clear. What is the point the authors wish to make here?

* Page 20, line 479 and further, the authors stress the importance of weight loss interventions and subsequent health gains. Indeed, efficacy studies show weight loss is possible in AP users, but studies carried out in practice show weight loss is very hard to achieve when patients are already overweight or obese. I would suggest the authors stress weight gain prevention by closely monitoring weight when starting/switching AP, supporting patients to eat healthy and remain/become active. Losing (a lot of excess) weight is much more difficult than not gaining it.

* Page 21, line 513, authors mention they are confident Chinese studies are methodologically sound, while other meta-analyses excluded Chinese studies for methodological reasons. Could the authors explain why they are confident about methodological soundness while others are not?

* Page 22, line 526, authors are worried about studies with null-findings not being studied. But did authors check trial registries?

* page 22, line 542, the authors state "The AP-naive AIWG was noticeable for all antipsychotics in short and

longer terms, triggered by APs induced increase of appetite." This latter part, the weight gain being triggered by APs induced increase of appetite, was not topic of this study and should not be mentioned in the conclusion. This is not what this study found (nor is it found in other literature; surely the increased appetite plays a role but this is not the only cause of AIWG).

* Page 23, line 549, again, besides (or rather than) stressing the importance of losing body weight by lifestyle counseling, clinicians should really try to avoid weight gain by offering this counseling when patients start using AP.

Minor remarks:

* I have read several times: "increased bodyweight gain" but I do not think this is right. Shouldn't in be either increased body weight, or body weight gain.

*page 5, line 118, the word 'that' does not seem right in this sentence.

* Page 6, line 138/139, this sentence is not correctly formulated.

*Page 10, line 231/232, this sentence does not read well, it seems some words are missing.

Page 13, line 303, increase in weight gain?

* Page 15, line 361, indicating AP switch do not differ compared.... This sentence is difficult to understand (does not seem correct)

* Page 18, line 448, result should be results.

* Page 20, line 490, underestimation is one word.

* Page 21, line 512, are should be were.

* Page 22, line 530/531, this sentence is not complete and not easy to understand.

* Page 23, line 551/552, references are not correct

* In the tables, often , is used rather than . for the decimals.

Reviewer #2: This is a fairly well done paper on weight gain with antipsychotics. Its advantage is that it separates weight gain into various lengths of treatment, compares weight gain in naive subject’s vs switch subjects, and tries to see if there are any diagnostic differens in weight gain effects of antipsychotics. It is a more traditional metaanalysis and not a network metaanalysis, so it cannot as precisely compare different antipsychotics with all the possible intersecting data.

I believe several clarifications or additions would add to the value of clarify the import of the paper

1) the English is not perfect and needs correction.

There are many places where the language is stilted or non-colloquial English and there are grammatical or sentence structures error. For example, in many places trying to refer to several antipsychotics and site it as AP not APs. Some of the language is stilted.

In lines 232-234 it is not that clear what you are saying “in the paper not presented in original paper” “this instance” etc. lines 377 etc. indicate that, doesn’t really start a new grammatical sentence.

There are several other similar errors and stilted language. Maybe a native English writer should go over the paper and correct.

2) Figure legends are usually presented as a separate file, not internally in the text, and then the final print version put each figure legend below the figure.

3) you say you selected APs where there was at least information for 2 studies, (LINE 257), but tables 1 and 2 show many entries for an antipsychotic at a specific longest length of treatment where there was only one study, so I think you have to clarify this, or you expressed it incorrectly in the English.

4) Both in the tables and figures or legends to the tables and figures you mix up N and n. You seem to say N is for number for studies and n is for number of subjects, but then in legends to the tables or the bottom of the figures you have only N and N, so things are not clear.

5) It is very hard to read the base of figure 3. It is very hard to make out the words and numbers. In Figure 2 things are larger and so can be read although there is still confusion between number of studies and number of subjects because both are in capital N. Can you put the bottom of table 3 in bold, or make the font bigger, so things can be read?

6) Please specify in the legends to tables 1 and 2 how the Es was calculated. Is this mean difference, standardized mean difference or what?

7) The results and discussion state that the supposed more weight friendly APs—such as amisulpride, aripiprazole and ziprasidone, were not associated with significant overall weight loss in switch studies. It would help clarify if these switch studies included switches from many different antipsychotics, or only or mostly from olanzapine or clozapine. If one were to restrict the switch studies to switches from olanzapine or clozapine, are there enough studies to make a conclusion, and would these studies show a weight loss when switching to one of these three alternative antipsychotics.

8) Maybe the authors have to caution about switching in terms of balancing efficacy and wright gain and side effects. Metaanalyses has generally shown clozapine to be the most efficacious antipsychotic especially for treatment resistant patients, and a few studies show olanzapine may be effective for some treatment resistant patients. In some metaanalyses olanzapine is ranked 2 or 3rd as most efficacious, although some of the differences are not large. So maybe there should be a cautionary comment about weight gain adverse effects vs efficacy, especially since some studies with olanzapine and clozapine show that clinical response was correlated with weight gain.

6. PLOS authors have the option to publish the peer review history of their article (what does this mean?). If published, this will include your full peer review and any attached files.

Reviewer #1: **Yes: **Frederike Jörg

Reviewer #2: **Yes: **Robert C Smith MD

---

## [Author Response · Author response to Decision Letter 0]

9 Dec 2020

Dear reviewers.

Thank you for your thorough and most helpful comments. It was appreaciated. We worked hard to answer your comments. We feel it had improved the manuscript and we may have clarify some of the results and discussion.

---

## [Decision Letter · Decision Letter 1]

21 Dec 2020

Antipsychotics results in more weight gain in antipsychotic naive patients than in patients after antipsychotic switch and weight gain is irrespective of psychiatric diagnosis: a meta-analysis

PONE-D-20-24708R1

Dear Dr. Bak,

We’re pleased to inform you that your manuscript has been judged scientifically suitable for publication and will be formally accepted for publication once it meets all outstanding technical requirements.

Kind regards,

Kyle J Burghardt

Academic Editor

PLOS ONE

Reviewers' comments:

Reviewer's Responses to Questions

**Comments to the Author**

1. If the authors have adequately addressed your comments raised in a previous round of review and you feel that this manuscript is now acceptable for publication, you may indicate that here to bypass the “Comments to the Author” section, enter your conflict of interest statement in the “Confidential to Editor” section, and submit your "Accept" recommendation.

Reviewer #1: All comments have been addressed

Reviewer #2: (No Response)

2. Is the manuscript technically sound, and do the data support the conclusions?

Reviewer #1: Yes

Reviewer #2: Yes

3. Has the statistical analysis been performed appropriately and rigorously? 

Reviewer #1: Yes

Reviewer #2: Yes

4. Have the authors made all data underlying the findings in their manuscript fully available?

Reviewer #1: Yes

Reviewer #2: Yes

5. Is the manuscript presented in an intelligible fashion and written in standard English?

Reviewer #1: No

Reviewer #2: Yes

6. Review Comments to the Author

Reviewer #1: The authors have indeed addressed all comments sufficiently and revised the manuscript adequately.

However, I would recommend a thorough review by a native English speaker as there are still quite a few mispellings (already in the title: "antipsychotics results" (antipsychotics is plural so no s behind result).

Reviewer #2: (No Response)

7. PLOS authors have the option to publish the peer review history of their article (what does this mean?). If published, this will include your full peer review and any attached files.

Reviewer #1: **Yes: **Frederike Jörg

Reviewer #2: No

---

## [Editor Report · Acceptance letter]

2 Feb 2021

PONE-D-20-24708R1 

Antipsychotics results in more weight gain in antipsychotic naive patients than in patients after antipsychotic switch and weight gain is irrespective of psychiatric diagnosis: a meta-analysis 

Dear Dr. Bak:

I'm pleased to inform you that your manuscript has been deemed suitable for publication in PLOS ONE. Congratulations! Your manuscript is now with our production department. 

Kind regards, 

on behalf of

Dr. Kyle J Burghardt 

Academic Editor

PLOS ONE